# Maxillary Antral Pseudocyst Drift after Osteotome Sinus Floor Elevation with Simultaneous Implant Placement: A Case Report and Literature Review

**DOI:** 10.3390/jcm12030920

**Published:** 2023-01-24

**Authors:** Peihan Wang, Nan Huang, Jiayin Ren, Ping Gong, Jie Long, Bo Huang

**Affiliations:** 1West China School of Stomatology, Sichuan University, Chengdu 610041, China; 2State Key Laboratory of Oral Diseases, National Center of Stomatology and General Dentistry, West China Hospital of Stomatology, Sichuan University, Chengdu 610041, China

**Keywords:** maxillary antral pseudocyst, drift behavior, implant rehabilitation, osteotome sinus floor elevation

## Abstract

This report describes maxillary antral pseudocyst drift after maxillary sinus floor augmentation through osteotome sinus floor elevation with simultaneous implant placement. 3D Slicer was used to measure the pseudocyst and maxilla for the placement of the implants; follow-up visits were scheduled at 6, 12, and 22 months. No adverse effects were observed during or after surgery, and all implants exhibited osseointegration without mobility. At 6 months after surgery, the pseudocyst had moved posterolaterally from the preoperative position near the anterior medial maxillary sinus, then returned to its original position at 12 months. However, it had remigrated to the posterolateral position at 22 months. The preoperative volume of the pseudocyst was 3.795 mm^3^; it was 2.370, 3.439, and 2.930 mm^3^ at 6, 12, and 22 months after surgery, respectively. The changes in pseudocyst drift and volume did not have a substantial negative influence on the implants, presumably because of cystic attachment and the recurrence of multiple pseudocysts at different locations. The risks associated with changes in a pseudocyst can be avoided, if an appropriate treatment plan is selected.

## 1. Introduction

Anatomical variations and lesions of the maxillary sinus are common findings in cone beam computed tomography (CBCT) scans of the maxilla prior to placement of dental implants [1]. A clinical study of oral implant patients in a Chinese grade A stomatology hospital showed that the prevalence of maxillary antral cysts was 11.41%; many of these were pseudocysts, with dome-shaped soft tissue findings in radiographic assessment of the maxillary sinus floor [2,3].

In our clinical experience, relevant clinical symptoms are not present in most patients who exhibit a maxillary antral pseudocyst during radiographic assessment prior to implantation. Therefore, many patients prefer a conservative approach (i.e., pseudocyst preservation during implant surgery) because they have concerns about the risk of pseudocyst management surgery. Although no standard treatments and implant options have been established for patients with such antral lesions, the current literature suggests that the presence of a pseudocyst should not constitute an absolute contraindication for sinus grafting. Notably, surgery using a maxillary sinus lift procedure to resolve bone mass deficiency, which has achieved a generally high success rate in terms of guided bone augmentation and implant survival; the presence of a pseudocyst in the maxillary sinus may not influence the clinical effects of raising the sinus floor during healing [3,4,5].

A recent study showed that dental implant placement after antral augmentation in patients with pseudocysts is safe and results in a high survival rate, regardless of pseudocyst removal or preservation [6]. Although complete removal is the gold standard for the management of cystic lesions, this approach is limited by high rates of injury and complications [7,8]. Some studies have suggested that, unless clinically significant expansion is evident during radiographic assessment, or symptoms (e.g., headache) are present, a “wait and see” approach is an appropriate strategy for the management of retention cysts or pseudocysts [9,10,11,12,13].

The current literature contains multiple studies regarding the management of maxillary sinus pseudocysts associated with implantation, with a focus on the impacts of different management approaches; however, there are few reports of intraoperative and postoperative changes in these pseudocysts [3,6,8,9,14,15,16]. Notably, implant status is a key consideration during postoperative follow-up; however, because many patients refuse clinical management of asymptomatic pseudocysts, considerations regarding such pseudocysts should not be limited to preservation or surgical removal. Changes in pseudocyst size, location, and morphological characteristics after implantation should be closely monitored to guide subsequent treatment and facilitate risk assessment.

Since 1944, when Summer introduced his technique, osteotome sinus floor elevation using a trans-alveolar approach [17], the trans-alveolar approach for sinus floor elevation has been a recommended approach to address insufficient vertical bone height. This classical method can not only increase the density of the posterior maxillary, but also promote the initial stability of the implant [17,18]. Liu et al. [18] reported that immediate implant placement, combined with maxillary sinus floor elevation utilizing the trans-alveolar approach, revealed a satisfactory clinical effect in both submerged healing and non-submerged healing for the maxillary molar area.

Here, we describe implant placement with simultaneous maxillary sinus floor augmentation via osteotome sinus floor elevation in the presence of an antral pseudocyst. During the perioperative period and postoperative follow-up, the maxillary sinus pseudocyst exhibited recurrent drift on the implant side.

## 2. Case Report

A 53-year-old woman was referred to the Department of Oral Implantology, West China Stomatology Hospital of Sichuan University, for implant rehabilitation because of the bilateral loss of maxillary premolars and molars.

### 2.1. Preoperative Examinations

The gingiva surrounding the absent maxillary posterior teeth were in good condition, without ulceration or swelling. The patient consented to surgical treatment, with an initial focus on the left maxillary posterior region.

The anterior teeth of the patient were of shallow overjet and shallow overbite, and the occlusal relationship was relatively stable. The gingiva, with a normal width of attached gingiva, belonged to the biological type of thin gingiva. The patient’s temporomandibular joint was in a good condition, and the opening degree was greater than 3 fingers. The patient had no obvious symptoms of maxillary sinus blockage, runny nose, snoring, or and nasal congestion, did not exhibit any bad habits, such as night bruxism and one-sided chewing. No systemic diseases, such as diabetes, heart disease, or infectious disease, nor history of trauma or surgery, was reported.

CBCT examinations were performed on CBCT scanners (3D Accuitomo, J. Morita Mfg. Corp., Kyoto, Japan) using the following parameters: 85-kV tube voltage, 5.0-mA tube current, field of view of 100 mm × 50 mm, and slice thickness of 1.0 mm. Dental CBCT scans showed that the local alveolar ridge width was optimal (8.688 mm for the left maxillary second premolar and 10.130 mm for the first molar) (Figure 1B,D); however, sinus augmentation was needed because the height of the residual alveolar bone in the posterior maxillary area was insufficient for implant placement. The patient was observed to have type 3 bone quality in the posterior maxilla on both sides, and according to our observation, there was no obvious thickening of the nasal mucosa in the pre-operative CBCT image. Notably, the height of the crestal bone between the sinus floor and the alveolar ridges of the left maxillary second premolar and first molar were 8.688 mm and 4.385 mm, respectively (Figure 1A,C). CBCT also revealed a homogeneously opaque, dome-shaped, and well-delineated lesion (volume, 3.795 mm^3^) in the left maxillary sinus (Figure 1 and Figure 2A); the lesion did not exhibit an epithelial lining. Prior to attending our clinic, the patient had consulted an otolaryngologist and received confirmation that no mucous cyst was present (no aggressive lesion appearance, significant bone resorption, or invasion of adjacent structures observed on radiologic imaging). Because the patient did not exhibit relevant symptoms, the otolaryngologist suggested conservative management of the pseudocyst. Accordingly, the treatment plan comprised osteotome sinus floor elevation, with simultaneous implant placement on the left side, in the presence of an antral pseudocyst.

### 2.2. Surgical Procedures and Postoperative Management

Preoperative hematological tests revealed values within normal limits, and the patient did not show other systemic abnormalities. Thus, local anesthesia was induced with articaine hydrochloride plus 1:100,000 adrenalin (Primacaine, Merignac, France).

Osteotome sinus floor elevation was performed in the left posterior maxilla, followed by implantation of bovine bone xenograft (0.25 g; Bio Oss, Wolhusen, Switzerland). Two ITI implants (4.8 × 8 mm and 4.1 × 10 mm, ITI, Straumann, Basel, Switzerland) were inserted in the augmented sinus in a submerged mode. The torque values were 30 N·cm and 15 N·cm, respectively. During surgery, there was no pseudocyst rupture, pseudocyst fluid exudation, or maxillary sinus membrane perforation. Optimal implant orientation and spacing were achieved. Bone compression revealed acceptable initial stability; thus, healing abutments were simultaneously connected. The tissue flap was finely sutured, and no adverse effects were observed during or after surgery.

The patient was instructed to take the following medications after surgery: amoxicillin (Zhongnuo, Shijiazhuang, China), 1.5 g/day for 3 days; ornidazole dispersible (Meheco Topfond, Zhumadian, China), 1.0 g/day for 3 days; dexamethasone acetate (Xianju, Taizhou, China), 3 mg/day for 3 days; diclofenac sodium (Simcere, Nanjing, China), 0.1 g/day for 3 days; and compound gargle solution chlorhexidine gluconate (Chenpai Bond, Haimen, China), 3 times daily for 3 days. Routine follow-up visits were scheduled at 6, 12, and 22 months.

### 2.3. Postoperative Examinations

Although the final crown restoration (second procedure) was planned for 6 months after the first procedure, it was delayed until 10 months because of an outbreak of COVID-19. A CBCT scan at the 6-month follow-up showed that the implants had been correctly placed in the augmented sinus, resulting in osseointegration without mobility (Figure 3B). CBCT scans at 12 months and 22 months showed stable bone levels around the implants (Figure 3C,D). The patient reported no discomfort or other complaints in relation to the implants or the maxillary sinus, and no fluid leakage was evident on the incision line during follow-up. However, a comparison of the preoperative CBCT scans with the postoperative scans conducted at 6, 12, and 22 months revealed pseudocyst drift in the left maxillary sinus (Figure 4). At 6 months after surgery, the pseudocyst had moved posterolaterally from the preoperative position near the anterior medial maxillary sinus, then returned to its original position at 12 months. However, it had remigrated to the posterolateral position at 22 months.

The pseudocyst volume was manually segmented using 3D Slicer and corrected by a professional radiologist [19]. Measurements of the maxillary sinus pseudocyst revealed that its volume decreased from 3.795 mm^3^ before surgery to 2.370 mm^3^ at 6 months post-surgery; it gradually returned to the preoperative volume at 12 months (3.439 mm^3^), then decreased to 2.930 mm^3^ at 22 months (Figure 2).

## 3. Discussion

In this report, we have described the clinical treatment of a patient with a pseudocyst who underwent implant surgery, along with the postoperative findings that included an intriguing type of pseudocyst drift not reported in previous literature. Postoperative imaging showed that the presence of the pseudocyst, as well as its drift behavior, did not have a substantial adverse effect on the clinical outcomes of maxillary sinus augmentation and implantation. Successful osseointegration and stable bone levels around the implants were observed during follow-up.

Pseudocyst drift may result from the following factors. First, the pseudocyst may not be completely immobile on the maxillary sinus floor. The pseudocyst may be entirely detached from the sinus floor, or it may be solely attached to the anterior portion of the sinus floor. In either situation, various forces, including implant insertion, airflow pressure from breathing, and postural changes during radiographic assessment could influence pseudocyst location. Second, considering the changes in volume, multiple pseudocysts may be present; the sizes of the pseudocysts may increase and decrease over time.

The apparent reduction in pseudocyst volume after implantation may have resulted from the leakage of pseudocyst fluid during implantation; however, no obvious pseudocyst fluid leakage was observed during surgery because of difficulty distinguishing such fluid from saline solution. However, follow-up imaging revealed re-enlargement and re-contraction of the pseudocyst, indicating that the changes in volume were not solely related to surgery. We suspect that changes in the secretion and absorption of fluid from a single pseudocyst contributed to these imaging findings; alternatively, multiple pseudocysts were present in our patient. If multiple pseudocysts are present, the changes in volume may represent different pseudocysts and could explain the apparent pseudocyst drift behavior mentioned above.

Mucosal cysts of the maxillary sinus are common benign lesions that are generally harmless and asymptomatic; the incidence of such lesions ranges from 1.6% to 11.41% [2,14,15]. These lesions have three forms that are classified on the basis of their clinical features and biological characteristics: (1) true mucous cysts—thick-walled cystic lesions that can destroy bone and expand into adjacent structures, usually identified by their radiographic appearance; (2) retention cysts—small, opaque, mucus-filled lesions caused by the obstruction of mucosal glands, usually located near seromucinous glands around the ostium; (3) pseudocysts—lesions attached to the floor of the maxillary sinus, typically dome-shaped without an epithelial lining and occasionally filling the entire antral cavity, that either persist unchanged or disappear without an apparent reason [3,13,20] No medical or surgical treatment is needed for most mucosal cysts of the maxillary sinus if they exhibit spontaneous regression or maintain a consistent size [15].

Because there is no consensus regarding the indications and standard approach for sinus augmentation in patients with maxillary antral pseudocysts, multiple studies have investigated the clinical outcomes of available treatment options. We summarize the principal progress in recent research on this topic (Table 1). 

Fu et al. [21] conducted a retrospective cohort study, dividing 26 patients into two groups (13 in the “removing the cyst group” and 13 in the “leaving the cyst alone” group). The results showed that antral pseudocyst removal before maxillary sinus floor augmentation and immediate implant placement after lateral sinus floor elevation could achieve higher bone graft volumetric stability. However, another retrospective cohort study revealed that the transcrestal raising of the sinus floor in the presence of antral pseudocysts may not have any influence on the clinical effects of raising the sinus floor during healing [5]. Sinus augmentation through a transcrestal window approach was described as a minimally invasive method for maxillary sinus augmentation [30]. Chiapasco and Palombo [24] combined pseudocyst enucleation with sinus grafting through a small bony window, which resulted in the survival of all 17 implants in their study. In addition to the discussion on whether to or not remove antral pseudocysts, researchers have also been focusing on improving the technology or combine the traditional method with some new technologies to reduce patient pain and various complications. Oh et al. [23] provided a technique combining the aspiration of the cysts during sinus floor elevation with a sinus augmentation procedure using a hydraulic sinus elevation system with simultaneous placement. After 6 months, a significant amount of bone formation around the implant at the sinus floor was observed, without evidence of cyst recurrence. Hu et al. [22] believed that the transoral endoscopic technique could be of great help in sinus floor augmentation, benefiting from its minimal invasion and optimal visualization. They successfully used the endoscopic-assisted intraoral surgical technique for sinus floor augmentation and the simultaneous removal of an antral pseudocyst. This method not only provided a clearer field of view for the complete removal of pseudocysts, but was also sensitive to small perforation, and only a small window was needed to observe the occurrence of mucosal perforation. Liu et al. [14] reported simultaneous implant placement with maxillary sinus elevation, along with pseudocyst treatment via cystic fluid extraction; all 14 patients were successfully treated without any discomfort, and all 28 implants were stable after 1 year of follow-up. Tang et al. [16] reported successful sinus augmentation without pseudocyst removal, along with simultaneous placement of two implants on the pseudocyst side; 1 year later, they observed a reduction in the dome-shaped opacity without bone graft resorption.

In patients with pseudocysts, the antral ostium may be blocked by obliteration of the sinus cavity, if bone graft overfilling occurs during sinus augmentation [4,6,7,8]. In addition to antral floor elevation using a lateral approach, the combination of an irregular graft surface with conventional bone graft volume may cause excessive pressure on the pseudocyst, leading to postoperative drainage of the pseudocyst fluid [3]. In our case, osteotome sinus floor elevation was chosen to minimize mucosal stimulation, antral sinus membrane perforation, and the loss of maxillary sinus space. This method involved using the concave edge of the internal lifting instrument to push the autologous bone in a manner that facilitated the formation of new bone in the sinus cavity. Moreover, the bone extrusion helped to improve the initial implant stability. The achievement of a satisfactory and stable clinical outcome, as well as the maintenance of pseudocyst integrity, indicated that our approach can reduce the risk of postoperative complications.

The cause of pseudocyst drift merits further investigation. However, our clinical findings may provide insights for other dentists. Notably, pseudocyst drift did not adversely affect the clinical outcomes of implantation or the patient’s postoperative quality of life. Thus, our findings suggest that the presence of maxillary sinus pseudocysts should not be regarded as an absolute indication for implant surgery; some risks associated with changes in maxillary pseudocysts can be minimized or avoided if an appropriate treatment plan is selected.

## 4. Conclusions

Changes in pseudocyst drift and volume did not have a substantial negative influence on the implants; this phenomenon may be related to the status of cystic attachment and the recurrence of multiple pseudocysts at different locations. The risks associated with changes in the pseudocysts themselves can be avoided if an appropriate treatment plan is selected.

## Figures and Tables

**Figure 1 jcm-12-00920-f001:**
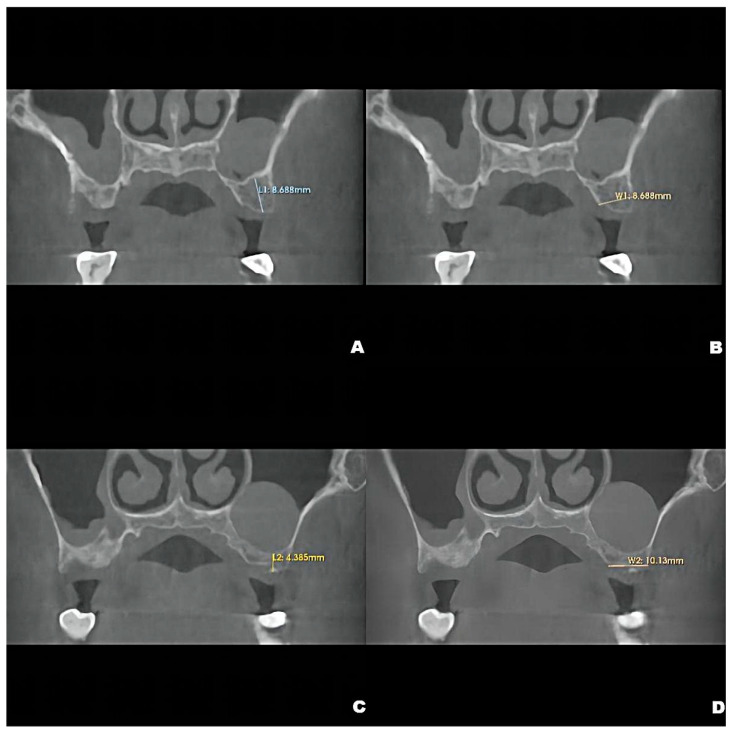
Preoperative CBCT showing the crestal height of L1 = 8.688 mm, (**A**), and local alveolar ridge width of W1 = 8.688 mm, (**B**), of the residual bone of the left maxillary second premolar and the crestal height of L2 = 4.285 mm, (**C**), and alveolar ridge width of W2 = 10.130 mm, (**D**), of the residual bone of the left maxillary first molar.

**Figure 2 jcm-12-00920-f002:**
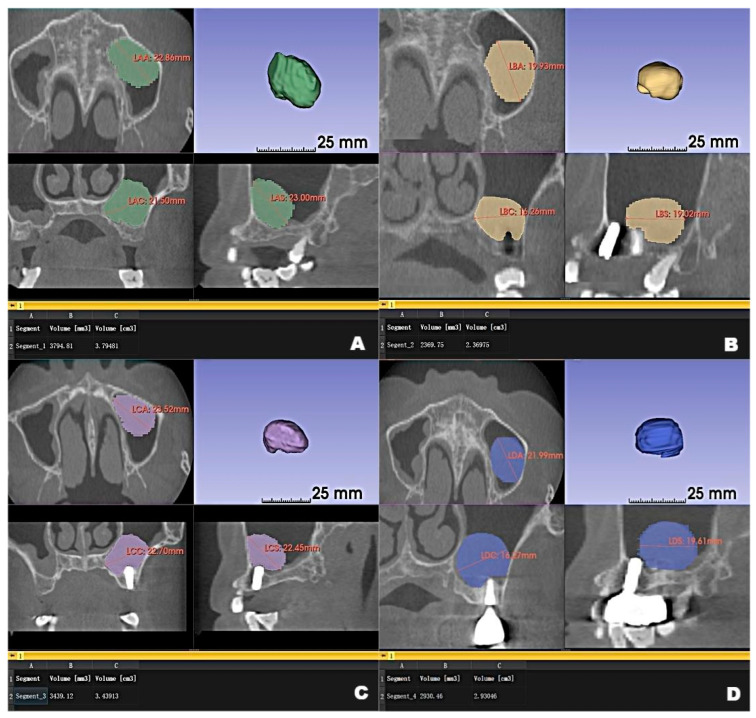
The volume changes and maximum diameter changes in the axial, coronal and sagittal planes of the patient’s left maxillary antral pseudocyst in the preoperative CBCT image (**A**) and the follow-up CBCT examinations in 6 (**B**), 12 (**C**) and 22 months (**D**), respectively.

**Figure 3 jcm-12-00920-f003:**
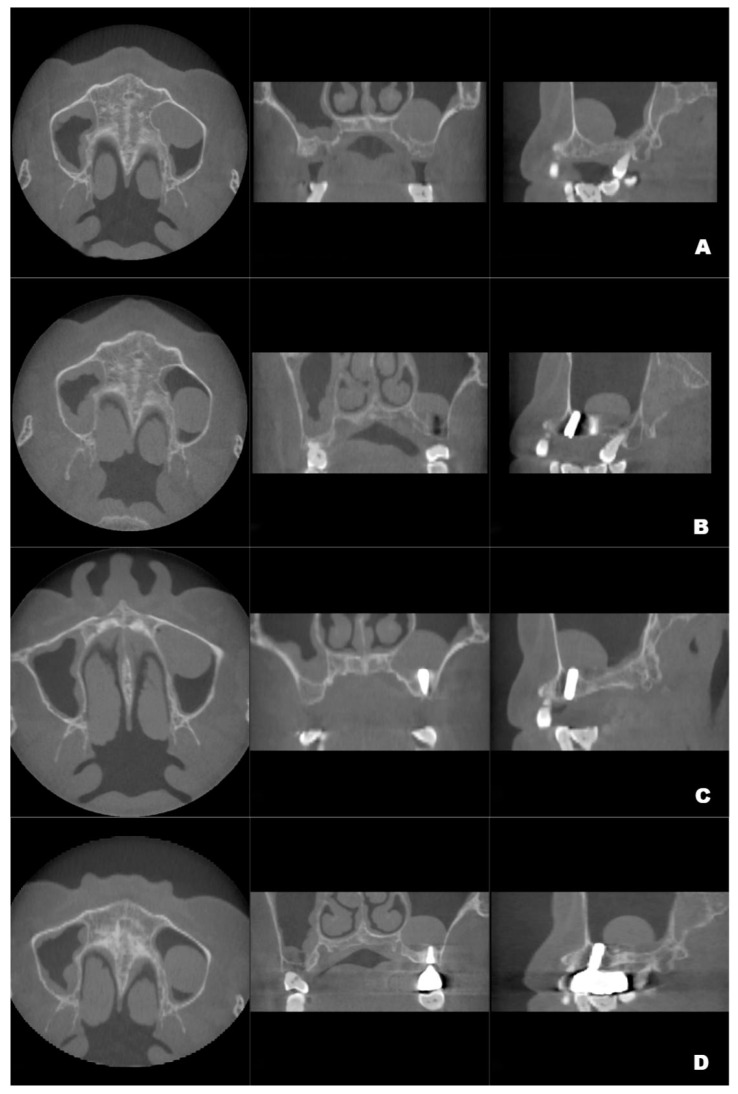
Preoperative CBCT (**A**) and follow-up CBCT images (6 months (**B**), 12 months (**C**), and 22 months (**D**)) of the patient’s left maxillary antral pseudocyst in the axial, coronal, and sagittal planes.

**Figure 4 jcm-12-00920-f004:**
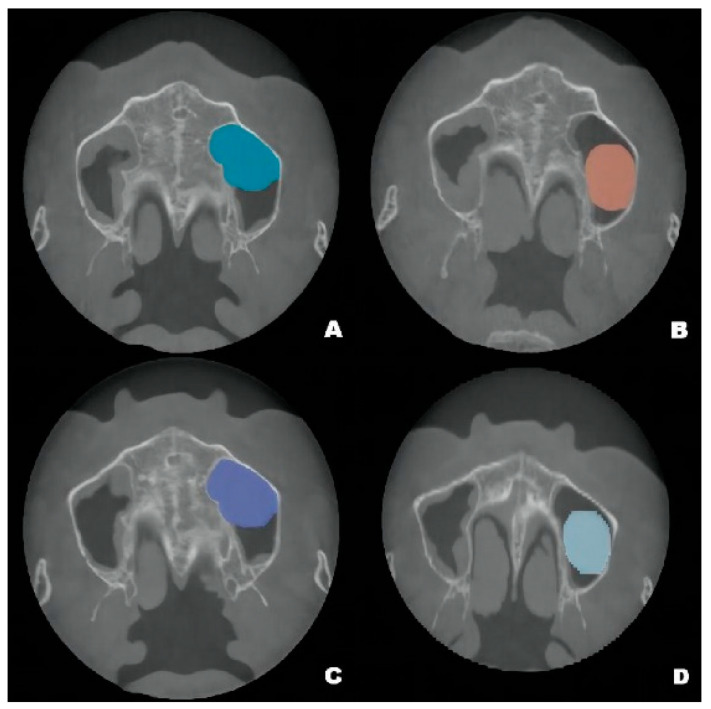
Pseudocyst drift behavior: the pseudocyst had moved posterolaterally from the preoperative position (**A**) at 6 months after surgery (**B**), then returned to its original position at 12 months after surgery (**C**). At 22 months, the pseudocyst had remigrated to the posterolateral position (**D**).

**Table 1 jcm-12-00920-t001:** Recent studies on the treatment options and corresponding clinical outcomes of maxillary antral pseudocysts.

NO	Author (Year)	Number of Cases	Surgical Method of Maxillary Antral Floor Elevation	Management of Antral Pseudocysts	Implant Timing	Follow-Up	Outcome
1	Fu et al. [21] (2022)	26	LSFE	Removed in 13 cases	1-Stage	12-month, ~2 to 5 years	Failure in 1 case at 12 months, with higher bone graft volumetric stability after 6–12 months.
				Preserved in 13 cases	1-Stage		Failure in 2 cases at 12 months and 2~5-years follow-up, respectively.
2	Gong et al. [5]	17	CSFE	Preserved	1-Stage	4–6 months	Osseointegration in all cases.
3	(2019) Yu and Qiu [8] (2019)	15	LSFE	Preserved	1-Stage	6 months	Failure in 1 implant before loading.
4	Liu et al. [14] (2018)	14	LSFE	Cystic fluid extraction	1-Stage	12 months	Osseointegration in all cases.
5	Hu et al. [22] (2017)	1	Endoscopic-Assisted LSFE	Removed	2-Stage	12 months	Osseointegration
6	Oh et al. [23] (2017)	2	CSFE	Cystic fluid extraction	1-Stage	6 months	Osseointegration in all cases.
7	Chiapasco and Palombo [24] (2015)	12	LSFE	Removed	1-Stage in 7 cases	12–96 months	Osseointegration in all cases.
					2-Stage in 5 cases		
8	Feng et al. [25] (2014)	21	OSFE	Preserved	1-Stage	Average of 27 months	Osseointegration in all cases.
9	Acocella et al. [26] (2012)	1	LSFE	Removed	2-Stage	2 years	Osseointegration
10	Cortes et al. [27] (2012)	1	LSFE	Removed	1-Stage	12 months	Osseointegration
11	Kara et al. [28] (2012)	29	LSFE in 17 cases	Preserved	1-Stage	Average of 17 months	Osseointegration in all cases.
			OSFE in 12 cases		2-Stage		
12	Celebi et al. [10] (2011)	4	LSFE in 2 cases	Preserved	1-Stage	6–8 months	Osseointegration in all cases.
			CSFE in 2 cases		2-Stage		
13	Tang et al. [16] (2011)	1	LSFE	Preserved	1-Stage	12 months	Osseointegration
14	Lin et al. [29] (2010)	11	LSFE	Removed	2-Stage	Average of 29.2 months after prosthetic loading	Osseointegration
15	Kara et al. [3] (2010)	2	LSFE in 1 case	Preserved	2-Stage	16 months	Osseointegration in all cases.
			CSFE in 1 case			13 months	
16	Mardinger et al. [9] (2007)	8	LSFE	Preserved	1-Stage in 7 cases	Average of 20 months	Osseointegration in all cases.
					2-Stage in 1 case		

LSFE: lateral sinus floor elevation; CSFE: crestal sinus floor elevation; OSFE: osteotome sinus floor elevation.

## Data Availability

The data are not publicly available to protect the information that contains the privacy of the participant. The data can be available from the author on reasonable request.

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
