# Peer review of "Maxillary Antral Pseudocyst Drift after Osteotome Sinus Floor Elevation with Simultaneous Implant Placement: A Case Report and Literature Review"

_jcm, 2023, doi:10.3390/jcm12030920_

Round 1

Reviewer 1 Report

The present paper discusses an interesting and much debated topic, that of maxillary sinus lesion management prior to implant placement and sinus lift procedures.

Please provide more recent references. Out of a total of nineteen references, only five are from the last 6 years.

Although the case is well presented, some issues are present and must be addressed, such as:

Please check reference format throughout the entire paper - ex: survival.[3, 4] - > survival [3,4].

Little information is provided regarding  local considerations such as patient's occlusion, biotype, bone density etc.

Also, more information about the maxillary sinus in this case must be provided such as antral ostium permeability prior to surgery, nasal mucosa state, nasal septum etc.

Please check statistics and quoted values (different in introduction and discussions):

A clinical study of oral implant patients in a Chinese Grade A Stomatology 30 hospital showed that the prevalence of maxillary antral cysts was 11.41%; many of these 31 were pseudocysts with dome-shaped soft tissue findings on radiographic assessment of 32 the maxillary sinus floor.[2, 3] and 

Mucosal cysts of the maxillary sinus are common benign lesions that are mainly 167 harmless and asymptomatic; the incidence of such lesions ranges from 1.6% to 9.2%.[13,14] 

Yet, the major concern regards the structure and content of the article. A systemic literature review would be welcomed before publishing a case report, in order to be able to have a relevant comparison between the presented technique and those already discussed in the published literature. 

Reviewer 2 Report

The case reported by the authors is clear and well-written.
Nevertheless, what they find is nothing special. The authors said “In this report, we have described the clinical treatment of a patient with a pseudocyst who underwent implant surgery, along with postoperative findings that included an intriguing type of pseudocyst drift not reported in previous literature” and conclude “Changes in pseudocyst drift and volume did not have a substantial negative influence on the implants”.

My main question is as follows:
What are the findings of your pseudocyst to be defined an intriguing type of pseudocyst?
I guess such findings are its changes of position and size over the time.
In my opinion it is nothing intriguing or new.

Minor concerns:

-       What kind of CBCT-unit did you use? How large was the field of view? What about technique parameters (kV, mA, etc…)?

-       In preoperative examinations, you said “Dental CBCT scans”. How many scans did you perform? One is usually enough.

-       You said “the patient had consulted an otolaryngologist and received confirmation that no mucous cyst was present”. How did the otolaryngologist confirm that no mucous cyst was present?

-       Figure 2 is too small.

Round 2

Reviewer 1 Report

The underlined issues were mostly addressed.

Reviewer 2 Report

Thanks for the changes made.

Author Response

Thank you for your comments and help!